# Ex-Vivo Measurement of the pH in Aqueous Humor Samples by a Tapered Fiber-Optic Sensor

**DOI:** 10.3390/s21155075

**Published:** 2021-07-27

**Authors:** Ondřej Podrazký, Jan Mrázek, Jana Proboštová, Soňa Vytykáčová, Ivan Kašík, Šárka Pitrová, Ali A. Jasim

**Affiliations:** 1Institute of Photonics and Electronics, Czech Academy of Sciences, 182 51 Prague, Czech Republic; mrazek@ufe.cz (J.M.); probostova@ufe.cz (J.P.); vytykacova@ufe.cz (S.V.); kasik@ufe.cz (I.K.); 2Clinic JL, 158 00 Prague, Czech Republic; pitrova@volny.cz; 3Faculty of Biomedical Engineering, Czech Technical University in Prague, 166 36 Prague, Czech Republic

**Keywords:** fiber-optic sensor, pH, HPTS, cataract surgery

## Abstract

A practical demonstration of pH measurement in real biological samples with an in-house developed fiber-optic pH sensor system is presented. The sensor uses 8-hydroxypyrene-1,3,6-trisulfonate (HPTS) fluorescent dye as the opto-chemical transducer. The dye is immobilized in a hybrid sol-gel matrix at the tip of a tapered optical fiber. We used 405 nm and 450 nm laser diodes for the dye excitation and a photomultiplier tube as a detector. The sensor was used for the measurement of pH in human aqueous humor samples during cataract surgery. Two groups of patients were tested, one underwent conventional phacoemulsification removal of the lens while the other was subjected to femtosecond laser assisted cataract surgery (FLACS). The precision of the measurement was ±0.04 pH units. The average pH of the aqueous humor of patients subjected to FLACS and those subjected to phacoemulsification were 7.24 ± 0.17 and 7.31 ± 0.20 respectively.

## 1. Introduction

A cataract is an eye disease where the lens becomes cloudy due to metabolic changes. This causes a gradual deterioration of vision, which becomes foggy or blurred, and leads to complete blindness if the affected lens is not removed. According to [1] the cataract is the leading cause of blindness and the second most frequent cause of severe vision impairment. The most common form is a senile cataract, affecting around 50% of people aged over 65 and 70% of those aged over 75.

The only cataract treatment known to date is based on surgical removal of the affected lens. The most commonly used method is so-called extracapsular extraction, carried out in local anesthesia. The operation removes only the core of the lens, which is first pulverized by an ultrasonic probe (so-called phacoemulsification) and then drained through a small incision made by a special scalpel. Recently, a gentler method called femtosecond laser-assisted cataract surgery can be used for both making cuts and fragmenting the lens core.

A prospective study of FLACS effects on the pH of human aqueous humor was performed by Rossi et al. in 2015 [2] to assess whether the plasma and cavitation bubbles, formed by the femtosecond laser, can affect the biochemical composition of the aqueous humor. They performed the measurements using a commercial pH-meter with a glass electrode, and found a significant difference of 0.9 pH units between the pH of the aqueous humor of patients subjected to phacoemulsification and those subjected to FLACS.

Glass electrodes are the gold standard in the measurement of pH thanks to their wide range, reliability, precision and affordable price. However, they are limited by their minimum size (diameter > 1 mm in case of miniaturized electrodes), rigidity and interference with electrical current and electromagnetic fields [3,4]. Optical fibers used as sensors offer, on the other side, some advantages in special applications where properties such as small size, flexibility, remote detection or resistance to the electromagnetic field are desirable [4]. Optical pH sensors are widely used in medicine, mostly in planar or fiber-optic arrangement. The first fiber-optic pH sensor, described more than 40 years ago by Peterson et al., was developed for pH measurement in blood [5]. Recently, applications of fiber-optic sensors were described for pH measurement in brain tissue [6], lungs [7,8,9], bladder and kidneys [10,11], oocytes [12], blood and subcutaneous tissue [13,14], breast tissue [15] and in lung cancer cells [16].

Despite numerous works dedicated to measurement of pH in various tissues and body fluids, there are only a few works dealing with the measurement of pH in human aqueous humor and all of them used pH electrodes for measurements [2,17,18,19].

Since the available sample volume of aqueous humor is small (typically 0.1–0.2 mL) and its pH value lies in a narrow physiological pH range between 6.5 and 7.5, a fiber-optic pH sensor is the ideal candidate for such a measurement. The typical optical sensors’ pH range of approximately three units is not limiting in this case, and the fiber tip diameter can be lower than 50 μm making the probe sufficiently small compared to the sample volume. The sensing layer thickness of a few microns provides short response times in the units of seconds [20].

A suitable optical pH sensor using a tapered fiber-optic probe has been developed in the Institute of Photonics and Electronics (IPE) since 2009, originally for the detection of pH in plant cells and tissues [21,22]. The design of the original sensor has been changed over time, as more suitable fiber-optic and optoelectronic components such as blue laser diodes and fiber-optic splitters have become available and the composition of the sensing layer was modified and optimized [23,24]. Recently, the functionality of pH-probes based on bioresorbable glass optical fibers was demonstrated [25,26].

Since the sensor has not undergone clinical testing yet, it could not be used for in-vivo measurements in human patients. The aim of the work was to design a method for ex-vivo measurement of aqueous humor pH and use it to compare two groups of patients subjected to phacoemulsification and FLACS respectively during cataract surgery.

## 2. Materials and Methods

### 2.1. A Fiber-Optic pH Sensor System

The pH sensor system is based on a hybrid sol-gel layer described by Wencel et al. [20], whose composition and way of preparation were modified concerning its stability at the tip of a tapered optical fiber [23].

Water-soluble pH-sensitive fluorescent dye HPTS is used as an opto-chemical transducer in ion-pairs with hexadecyltrimethylammonium bromide (CTAB), which makes the ion-pair lipophilic. HPTS exhibits two pH-dependent excitation bands centered at 405 nm and 465 nm wavelengths and a single fluorescence emission band centered at 508 nm. This makes it possible to take advantage of ratiometric fluorescence measurements by evaluating the sensor response, as the ratio of the emission intensity at 508 nm measured under excitation at 465 nm divided by the emission intensity measured under 405 nm excitation [20]. Undesirable effects, such as photobleaching, dye leaching or fiber bending, can be addressed by such an approach.

The HPTS-CTAB ion-pair is physically entrapped into a sol-gel matrix prepared from precursors (3-glycidoxypropyl)-trimethoxysilane (GLYMO) and ethyltriethoxysilane (ETES). GLYMO makes the matrix hydrophilic enough to be permeable for H_3_O^+^ ions, and ETES provides good adhesion and stability of the resulting layers. A detailed description of the sensitive layer preparation was published in [23].

The fiber-optic probe is made of a standard graded-index multimode silica optical fiber with its tip thermally tapered down to a diameter of approx. 15 μm. The tip is covered with the sensitive layer by dip-coating (Figure 1).

The probe is connected to an optoelectronic unit through a standard FC-PC optical connector. The optoelectronic unit provides excitation of the sensitive layer and detection of the fluorescence emission using two laser diodes at wavelengths of 405 nm and 450 nm, a photomultiplier tube (PMT) and a pair of fiber-optic couplers/splitters. The laser diodes are switched on and off in alternate manner and corresponding intensities are measured synchronously by PMT. The unwanted back-scattered excitation radiation is eliminated utilizing a bandpass optical filter, which passes the light in a range from 475 nm to 525 nm (Figure 2).

The sensor response is calculated as a ratio of the emission intensity values obtained for excitation at 450 nm and 405 nm:R = (I_450_ − I_amb_)/(I_405_ − I_amb_)(1)
where R is the calculated sensor response and I_405_ and I_450_ are the emission intensities measured at 405 nm and 450 nm, respectively. The intensity I_amb_, which is measured as the PMT output with both excitation laser diodes turned off, is subtracted from both fluorescence emission intensities I_405_ and I_450_ in order to diminish the influence of ambient light on the measurement.

The optoelectronic unit is connected to a computer with control software, which provides the setting of measurement parameters (time intervals, laser diode intensities, calibration, etc.), real-time data acquisition, calculation of the sensor response and of the pH value based on calibration, graphical representation of the pH value in time and logging of all the values into a file.

### 2.2. Characterization of the Sensing Layer

Spectral characterization of a prepared sensitive layer was performed using a fluorescence spectrometer (Fluorolog FL-3, Horiba-Jobin Yvon, Lille, France).

The sensitive layer was applied onto a fused silica glass planar substrate by spin-coating, and fluorescence excitation spectra were taken at an emission wavelength of 508 nm in a set of Britton and Robinson buffer solutions [27] in a pH range from 4.0 to 8.0. The emission spectra were taken in a buffer solution with pH = 6.0 at excitation wavelengths of 405 nm and 450 nm.

A calibration curve for sensor performance evaluation was measured with one tapered fiber-optic probe using the pH sensor system described above. A probe was dipped into flasks with Britton and Robinson buffer solutions in a pH range from 4.0 to 8.0. The arithmetic means of 30 values obtained after stabilization of the sensor response were used for calculation of the values used for the calibration curve.

### 2.3. Ex-Vivo Measurement of Aqueous Humor pH

A measurement cell was prepared by fixing an injection needle with a size of 25 G onto a plastic pad and inserting the fiber-optic probe inside the needle (Figure 3).

The cell was filled with buffer solution with pH = 7.0 at the beginning of each measurement; the measurement was started 5 to 10 min before the acquisition of the aqueous humor sample, and the response of the sensor was let to stabilize. The aqueous humor sample was taken into a 1 mL syringe during the operation just after the affected lens was removed (Figure 4). The content of the syringe was then injected into the injection needle of the measuring cell within 1 min after the sampling and subjected to pH measurement for several tens of seconds.

The excess sample flowing through the needle was collected into a waste reservoir formed by a spherical hole drilled in the base pad.

The remainder of the sample in the measuring cell was drawn back into the syringe at the end of the measurement, and the sensor response was measured in the same way using two buffer solutions with known pH values (7.0 and 7.5) lying above and below the typical physiological pH (7.3 to 7.4).

The arithmetic means of 30 values obtained after the sensor response had stabilized were used for the calculation. The pH of the sample was then calculated by linear interpolation between the responses measured for the buffer solutions:(2)$$pH=pH_{1}+\left(pH_{2}-pH_{1}\right)\frac{R-R_{1}}{R_{2}-R_{1}}\;$$
where *R* is the response measured for the sample, *R*_1_ and *R*_2_ are the sensor responses measured for the buffer solutions, and *pH*_1_ and *pH*_2_ are the pH values of the buffer solutions used for the calibration.

A total of 56 samples from 50 patients were analyzed (6 patients underwent cataract surgery on both eyes) with 28 samples taken after phacoemulsification and 28 samples taken after FLACS. The statistical analysis of obtained values was performed using Microsoft Excel statistical functions for two-sample *F*-test for variances and two-sample T-test assuming equal variances at a significance level of 0.05.

## 3. Results and Discussion

The spectral characterization of the sensitive layer spin-coated on a silica planar substrate is presented at Figure 5.

Two excitation peaks at 405 nm and 460 nm, whose maxima increase and decrease reciprocally with pH, and a single emission peak at 508 nm can be distinguished in the measured spectra. Since the sensor system uses 405 nm and 450 nm laser diodes for excitation, the values measured at those wavelengths were taken for constructing a calibration curve. A sigmoid curve defined by Boltzmann function was fitted through the points:(3)$$f\left(pH\right)=A_{2}+\frac{A_{1}-A_{2}}{1+e^{\frac{pH-pH_{0}}{\Delta pH}}}$$
where *A*_1_ and *A*_2_ are the upper and the lower limits of the calibration curve, *pH*_0_ is the inflection (central) point of the curve and Δ*pH* represents the maximal slope of the curve.

The resulting curve was compared to the curve obtained for the sensor system with the tapered fiber-optic probe to see how the relatively simple fiber-optic setup of the sensor system affects the pH measurement compared to the standard analytical instrument—the fluorescence spectrophotometer equipped with monochromators. A comparison of both curves after normalization is shown in Figure 6.

It can be seen that the normalized calibration curves are almost identical. The slight difference in their shape can be ascribed to different optical conditions, i.e., fluorescence spectrophotometer equipped with monochromators and a xenon arc lamp as the excitation source versus simpler setup of fiber-optic sensor system equipped with laser-diodes and a bandpass filter. Nevertheless, the inflection point of both curves is at the same point at pH = 5.85 ± 0.02. That value corresponds to the apparent acid dissociation constant (pK_a_) of the immobilized HPTS-CTAB ion-pair, and it is shifted by 1.45 units compared to the value reported for free HPTS in solution (7.30) [28] and by 0.43 units compared to the value reported for HPTS-CTAB ion-pairs in the original work of Wencel et al. (6.28) [20].

A total of 56 aqueous humor samples was measured over a three month period using 11 tapered fiber-optic probes. Since the typical pH of human aqueous humor lies in the range from 7.32 to 7.60 [2], the measurements were performed at the upper part of the calibration curve. Therefore, a two-point calibration was performed after each sample measurement with buffer solutions with the nearest lower and higher pH values to increase the accuracy of the measurement and to eliminate the negative influence of potential temperature changes and the sensor drift. An example of the sensor response (obtained from Equation (1)) in time during such a measurement is in Figure 7.

The response times of the sensor to changes in pH were shorter than one minute. Typically, four to six samples were measured with one probe over a period of 4 to 5 h, while the net time during which a probe was excited by the laser diodes was in a range from 10 to 20 min for each measurement (including stabilization before each measurement). Typical values measured with one probe in 6 different samples over a period of 4.5 h are given in Table 1.

R is the sensor response and σ is its standard deviation measured in a sample, R_1_, σ_1_ are the values measured in the buffer solution with pH = 7 and R_2_, σ_2_ are the values measured in the buffer solution with pH = 7.5.

The emission intensities I_450_ and I_405_ were measured in the buffer solution with pH = 7.0 during a set of 6 consecutive sample measurements with the same probe, and the corresponding sensor responses R were calculated (Table 2) to evaluate the sensor drift.

Although the intensities decreased by about 40% after 108 min of measurement, the decrease in sensor response (drift) was only 7.8%, being only 3.2% after the first 92 min. thanks to the ratiometric approach (Figure 8).

The calculated pH values in Table 1 were not affected by that decrease, because calibration was performed with buffer solutions after each sample measurement.

All the pH values were obtained with a precision of ± 0.04 or better. The average pH of the aqueous humor of patients subjected to FLACS was 7.24 ± 0.17. The average pH of the aqueous humor of patients subjected to phacoemulsification was 7.31 ± 0.20. The relative frequency distribution in 0.1 unit wide pH intervals for both groups is shown in Figure 9.

The statistical analysis revealed that the difference was not statistically significant between the pH of aqueous humor in a group of patients subjected to FLACS and that in the group subjected to phacoemulsification (*p*-value of 0.20). It is in contrast with results published in [2], which were measured with a compact pH meter with glass electrode. Since no further details were published about the way of sample measurement in [2], it is unclear whether that discrepancy originates from different sample handling, different sensing principle or relatively small groups of patients in both studies.

One of the challenges of measuring pH in small volumes is the large ratio between sample surface and volume. The pH of the sample may change due to the gas absorption when exposed to an atmosphere containing gases affecting the pH of aqueous solutions (carbon dioxide, sulfur dioxide, ammonia, etc.), or due to the sample evaporation. The larger the volume of the sample, the less serious the problem is as the surface increases with the square of its length while the volume increases with its cube. In our case, the sample was contained inside a syringe needle and the measurement took place approx. 15–20 mm from the needle tip so the contact of the sample with the atmosphere was minimized.

## 4. Conclusions

The presented work demonstrated the feasibility of using a fiber-optic sensor for the measurement of pH in real biological samples with small volumes, where sample handling and the measurement setup design are critical. Use of a ratiometric optical pH sensor with a tapered fiber-optic probe was successfully demonstrated for ex-vivo pH measurement in samples of human aqueous humor during cataract surgery. Samples were measured with precision and response times comparable to those of standard pH electrodes. The drift of the sensor by 3.2% after 92 min of net measurement time was compensated by calibration in two buffer solutions after each sample measurement.

## Figures and Tables

**Figure 1 sensors-21-05075-f001:**
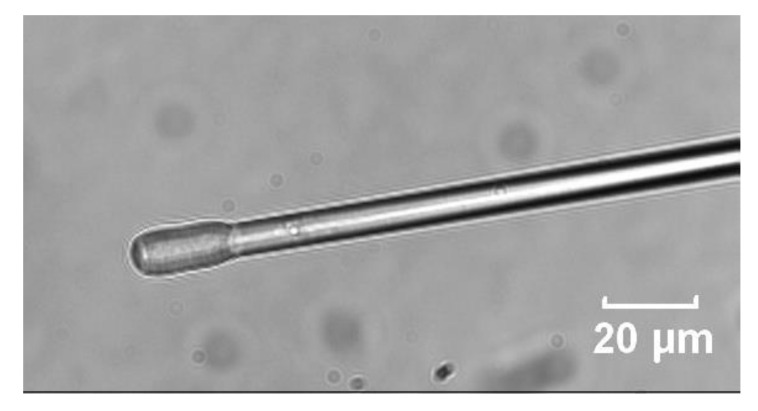
Microphotograph of the tip of a fiber-optic probe coated with the sensitive layer.

**Figure 2 sensors-21-05075-f002:**
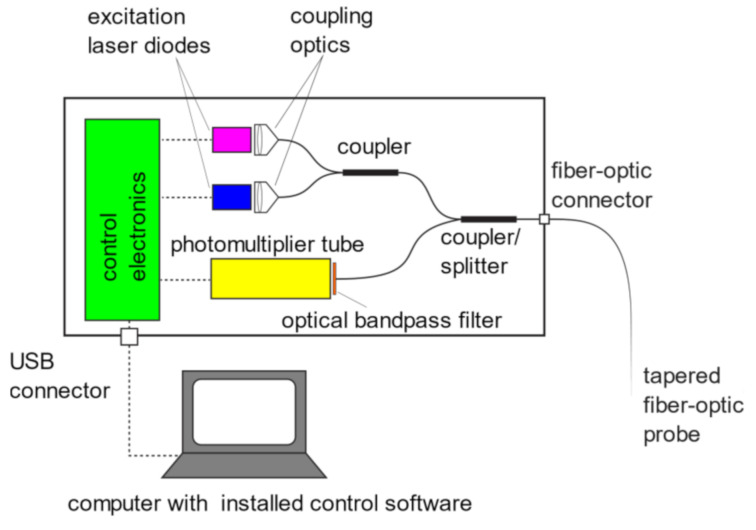
The scheme of the fiber-optic pH sensor system.

**Figure 3 sensors-21-05075-f003:**
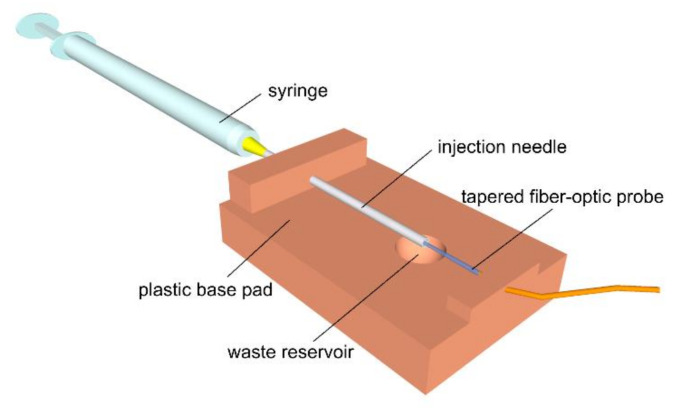
Diagram of the measuring cell.

**Figure 4 sensors-21-05075-f004:**
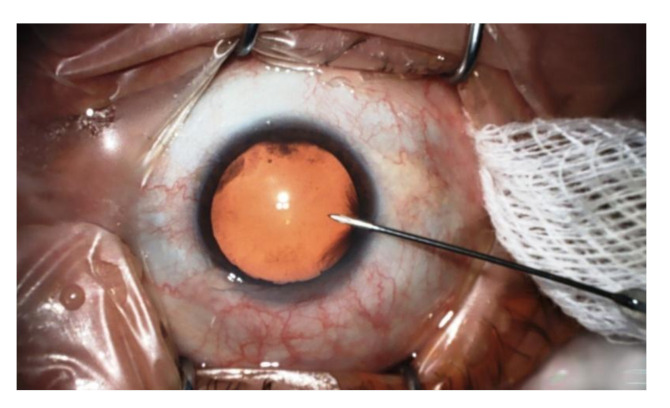
Retrieval of an aqueous humor sample during cataract surgery.

**Figure 5 sensors-21-05075-f005:**
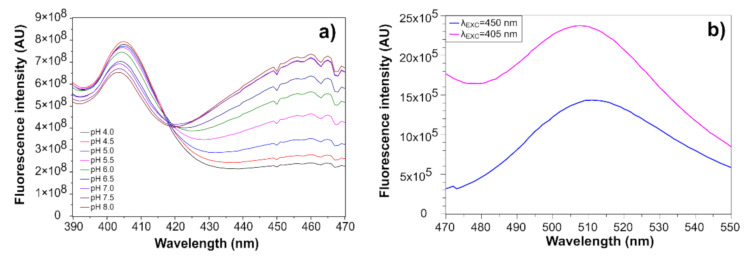
(**a**) Excitation spectra of the sensitive layer on a planar substrate taken at a 508 nm emission wavelength at different pHs; (**b**) and emission spectra taken at 405 nm and 450 nm excitation wavelengths in a buffer solution with pH = 6.0.

**Figure 6 sensors-21-05075-f006:**
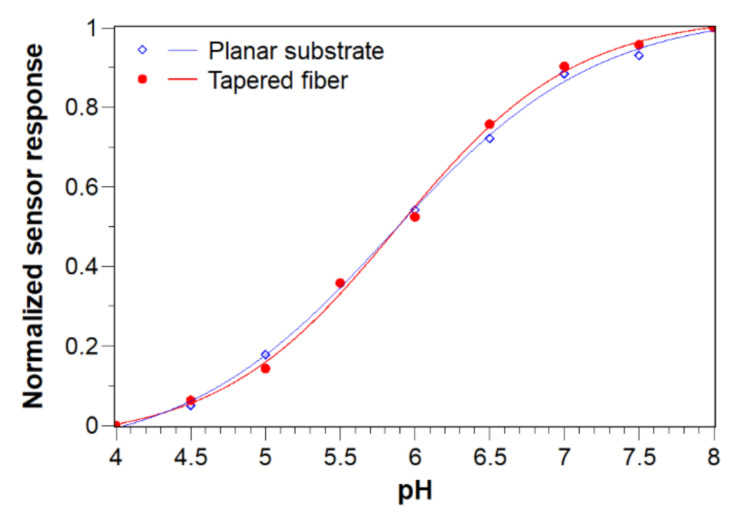
Normalized calibration curves measured on a planar substrate with fluorescence spectrometer (blue diamonds) and on a tapered fiber-optic probe with the sensor system (red dots).

**Figure 7 sensors-21-05075-f007:**
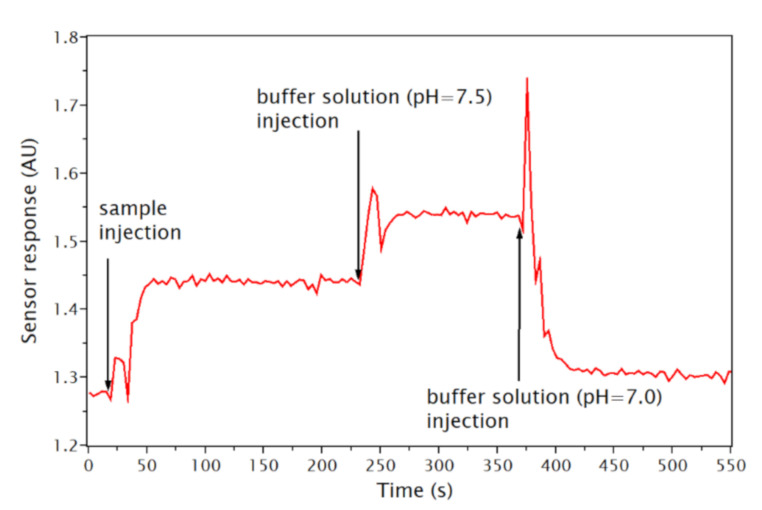
Sensor response in time during a measurement of a sample.

**Figure 8 sensors-21-05075-f008:**
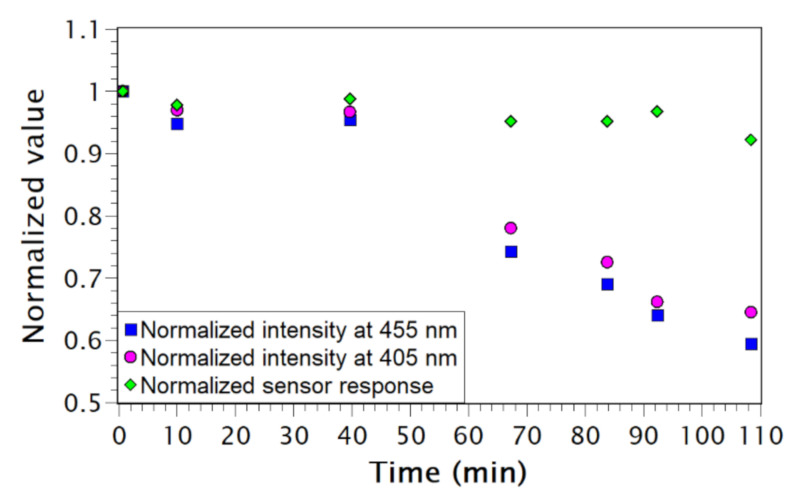
Decrease of the measured intensities and the sensor response for pH = 7.0 in time.

**Figure 9 sensors-21-05075-f009:**
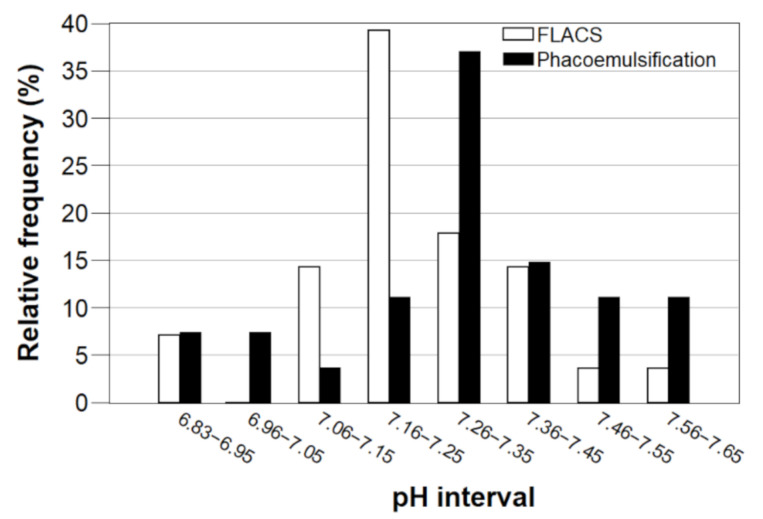
Relative frequency distribution of aqueous humor pH for FLACS and phacoemulsification.

**Table 1 sensors-21-05075-t001:** Overview of the sensor response and calculated pH values (σ-standard deviation; σ_r_-relative standard deviation).

Sample	Sensor Response						Calculated pH		
	R	σ	R_1_	σ_1_	R_2_	σ_2_	pH	σ_r_	σ_pH_
1	1.388	0.005	1.259	0.005	1.485	0.004	**7.24**	6.2%	**0.03**
2	1.517	0.010	1.348	0.008	1.603	0.004	**7.29**	8.3%	**0.04**
3	1.505	0.006	1.335	0.004	1.582	0.005	**7.30**	4.9%	**0.02**
4	1.449	0.005	1.300	0.004	1.558	0.005	**7.24**	5.0%	**0.02**
5	1.458	0.005	1.323	0.006	1.568	0.006	**7.23**	6.7%	**0.03**
6	1.437	0.004	1.301	0.004	1.535	0.004	**7.24**	4.8%	**0.02**

**Table 2 sensors-21-05075-t002:** Values measured in buffer solution with pH = 7.0.

Measurement	Meas. Duration (s)	Time from the Beginning (s)	I_450_	I_405_	R
Beginning	40	40	22,584	16,507	1.368
Sample 1	558	598	21,413	16,005	1.338
Sample 2	1778	2376	21,556	15,961	1.351
Sample 3	1657	4033	16,770	12,878	1.302
Sample 4	992	5025	15,591	11,974	1.302
Sample 5	514	5539	14,455	10,923	1.323
Sample 6	966	6505	13,430	10,647	1.261

## Data Availability

The data presented in this study are available on request from the corresponding author. The data are not publicly available due to ethical reasons.

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
