# Peer review of "Ex-Vivo Measurement of the pH in Aqueous Humor Samples by a Tapered Fiber-Optic Sensor"

_sensors, 2021, doi:10.3390/s21155075_

Round 1
Reviewer 1 Report
The authors have presented a fiber sensor for ex-vivo pH detection in aqueous humor samples. The sensing principle is based on the absorption difference of opto-chemical transducer at 405nm and 455nm. The idea is very interesting. However, the English writing is too rough. There are too many gramma and expressing issues, which is difficult to read and understand. I suggested to reject the current version and the authors should rewrite the manuscript.
- “The sensor uses 8-hydroxypyrene-1,3,6-trisul-10 fonate (HPTS) fluorescent dye as an Opto-chemical transducer, immobilized in a hybrid sol-gel matrix at the tip of a tapered optical fiber.” Not clear.
- “ Ex-vivo measurement of the pH in aqueous humor samples by 2 a tapered fiber-optic sensor.” There is period in the title.
- Please give the quantitative analysis results in the abstract,
- “if the lens is not removed” not the lens, should be affected lens.
- what kind of sol-gel used in the manuscript?
- Please give the detailed sensing principle.
- “The fiber-optic probe is made of a standard graded-index multimode silica optical 84 fiber with its tip thermally tapered down to a diameter of approx. 15 m. The tip is covered 85 with the sensitive layer by dip-coating (Figure 1Error! Reference source not found.).”
- 2 looks like from a patent, please add the name in the figure.
- There is not a definition to the parameter in equation (1)
- Fig 5 should include the the spectra information of 508nm.
- they are limited by their min-41 imum size
- “The presented work demonstrated the advantages of a fiber-optic sensor in the measurement of pH in real biological samples with small volumes,” the content presented in this manuscript is not the advantages.
Reviewer 2 Report
The paper presents a tapered fiber tip for ex vivo pH measurements of aqueous solutions. This is somehow a repetition of what authors have been doing with other published works where they measure the pH of different mediums. Also, the introduction must be done with relevant references of works developed by other groups rather than referencing the works of the authors. The introduction must be changed. I recall that in 19 references, at least 10 are from the authors. That is inappropriate. Relevant changes must be done for this work to be suited for Sensors.
I leave my comments about the developed work:
Calibration curve of the tapered fiber should be presented.
The full tapered section is sensitive to the external medium. From fig 1, it seems that the layer is only at the fiber tip. How do you ensure that only the tip is in contact with the solution?
It is not clear if intensity measurements are being done with the tapered fiber. Figure 7 – is intensity the sensor´s response (in arbitrary units)?
It is not clear the procedure explained in lines 136-139. Is the measured signal R retrieved from the sensor´s response presented in fig7? And about R1 and R2, what were the values used? This must be clarified, and an example of values should be given for better understanding.
It is not explained why a sensitive layer on a planar substrate is characterized (Fig 5) and why this should be used for the calibration curve when the calibration of the fiber sensor had already been done with calibrated pH solutions (line 114).
Reviewer 3 Report
This manuscript reported a pH measurement device with HPTS fluorescent dye as an optic system used for ex-vivo human aqueous humor detection. This sensor shows the advantages in real biological samples with small volumes. However, there are some questions need to be addressed.
1.Please notice the use of punctuation, such as Line 42–43, 185–186, 210–211; Table 1, Table 2.
- “The points were fitted by Boltzmann function:” I do not understand the mean of the function curve in here, please give explain in revision manuscript.
- Please give the represent message of every data in the detection calibration curve (Equation 3).
- Equation1: “R=(I445-Iamb)/ (I405-Iamb)” should be revised as “R=I445/I405”
- Figure 1, 2, and 3 should been combined into one figure. Figure 5, and 7 as same.
- Line 217–223: the part should be strengthened by adding supplement data and references.
- Please attention: I would kindly suggest authors for grammar check from a professional before submitting the manuscript.
Round 2
Reviewer 1 Report
Most of the issues the authors have corrected. However, there is still one more question.
I am very confused to the Excitation and Emission peak, please explain this in the manscript. what is the difference between them?
Author Response
We reformulated the explanation of HPTS fluorescence properties at lines 82-88. The reference no.20, where the principle is thoroughly described and which is cited at the line 77, was added again at line 87.
Reviewer 2 Report
The paper has been improved with the revisions. References have been updated. The paper is now suited for Sensors.
Author Response
Thank you again for the valuable comments.
Reviewer 3 Report
All points are fully addressed and revised by the authors, thereby no further corrections needed.
Author Response

(The authors gave the same response as above.)
